# Next-Generation Sequencing in Oncology—A Guiding Compass for Targeted Therapy and Emerging Applications

**DOI:** 10.3390/ijms26073123

**Published:** 2025-03-28

**Authors:** Laurenția Nicoleta Galeș, Mihai-Andrei Păun, Ioana Butnariu, Laurentiu Simion, Loredana Sabina Cornelia Manolescu, Oana Gabriela Trifănescu, Rodica Maricela Anghel

**Affiliations:** 1“Carol Davila” University of Medicine and Pharmacy, 020021 Bucharest, Romania; laurentia.gales@umfcd.ro (L.N.G.);; 2Department of Medical Oncology II, “Prof. Dr. Al. Trestioreanu” Institute of Oncology, 022328 Bucharest, Romania; 3Department of Neurology, National Institute of Neurology and Neurovascular Diseases, 077160 Bucharest, Romania; 4General Surgery and Surgical Oncology Department I, Bucharest Institute of Oncology “Prof. Dr. Al. Trestioreanu”, 022328 Bucharest, Romania; 5Clinical Laboratory of Medical Microbiology, “Marius Nasta” Institute of Pneumology, 050159 Bucharest, Romania; 6Department of Microbiology, Parasitology and Virology, Faculty of Midwives and Nursing, “Carol Davila” University of Medicine and Pharmacy, 020021 Bucharest, Romania; 7Department of Radiotherapy II, “Prof. Dr. Al. Trestioreanu” Institute of Oncology, 022328 Bucharest, Romania

**Keywords:** next-generation sequencing, targeted therapy, precision oncology

## Abstract

Multigene sequencing technologies provide a foundation for targeted therapy and precision oncology by identifying actionable alterations and enabling the development of treatments that substantially improve clinical outcomes. This review emphasizes the importance of having a molecular compass guiding treatment decision-making through the multitude of alterations and genetic mutations, showcasing why NGS plays a pivotal role in modern oncology.

## 1. Introduction

The unwavering pursuit of cancer cures has led to meaningful advances in multi-omic investigations, paving the way toward therapy tailored to the molecular intricacies of the cells. After successfully targeting the *BCR-ABL* rearrangement in chronic myeloid leukemia in 1998, the concept of *precision oncology* became pivotal for the development of therapeutic interventions tailored to the molecular specifics of individual tumors. Over 200 targeted therapies are currently approved, as of December 2024 [1], showing a growing interest in delivering the right treatment to the right patient at the right dose at the right time [2].

Consequently, precise diagnostic tools capable of accurately identifying treatment-impacting biomarkers have been developed since Next-Generation Sequencing (NGS) became almost synonymous with *precision oncology* and targeted therapy guidance. In essence, NGS enables a comprehensive genomic evaluation by parallel sequencing large quantities of DNA fragments extracted from tissue or liquid samples, resulting in a high-throughput, high-yield technique capable of interrogating sensitive molecular events with an impactful meaning in the subsequent clinical management [3]. Due to high specificity, NGS is evaluated for the development of multicancer early detection (MCED) tests [4].

NGS assays have received widespread institutional approval and are highly advocated in the current oncology guidelines on the pretreatment evaluation of a multitude of solid tumors [2] (Table 1).

Current National Comprehensive Cancer Network (NCCN), European Society for Medical Oncology, and American Society of Clinical Oncology (ASCO) guidelines recommend NGS in patients with advanced lung, breast, colorectal, prostate, and ovarian cancer, with indications ever expanding with the development of novel drugs [14,15]. Our current review is intended as a broad argument for the value of NGS in various clinical scenarios and the specific molecular alterations that can be interrogated and specifically targeted for the treatment of solid tumors.

## 2. The Current Role of NGS in Solid Tumor Oncology

### 2.1. Challenging Established Classifications and Redefining Diagnostics with Precision Oncology

Multi-omic research from past decades has provided sufficient evidence to prove the transformative role of certain molecular alterations concerning tumor biology, prognosis, and therapy. Such a paradigm shift occurred in the WHO classification of gliomas following the cIMPACT-NOW updates, which highlight the meaningful impact of simultaneously profiling key genes for an accurate diagnosis. Consequently, the 2021 EANO Guidelines for the diagnosis and management of gliomas showcase the importance of broad molecular profiling and correct identification of prognostic and predictive biomarkers [16,17]. We summarize the recommendations for the molecular-driven diagnosis and treatment of gliomas in Table 2.

Another example is the treatment for some rare tumor types, which has already proven the symbiosis between targeted therapy and gene sequencing for precise treatment administration. Gastrointestinal stromal tumors (GISTs) originate from the interstitial cells of Cajal and express a wide array of molecular alterations, including *KIT*, *PDGFRA*, and *SDH* alterations, providing different treatment options that require mutational testing for adequate treatment. Furthermore, mutations such as PDGFRA D842V confer insensitivity to well-established treatment agents such as Imatinib and, as such, require a broad investigation of all known molecular alterations associated with GISTs to recommend the precise therapy [18,19,20].

### 2.2. Guiding Compass for Tumor-Agnostic Tumors

The classical characterization of any cancer includes the location and tissue of origin as guiding elements for clinical management. With the possibility of interrogating for druggable molecular alterations, the concept of *tumor-agnostic therapy* has emerged as a new approach in oncology, emphasizing the specific driver of cancer growth as a treatment decider [21,22].

Although other methods exist for detecting aberrant mutations, such as FISH or RT-PCR, NGS is the only one that confers the advantage of a comprehensive genomic profile, including other targetable oncogenes. Furthermore, due to the high specificity and sensitivity, DNA NGS testing on formalin-fixed, paraffin-embedded (FFPE) tumor specimens is often used as the confirmatory test for FISH and RT-PCR.

In an encouraging report from Coquerelle et al., the researchers demonstrate improved access to clinical trials due to the better genetic profiling facilitated by NGS [23]. It is important to note that the pan-tumor paradigm shift was made possible through basket trials using NGS for patient inclusion, depending on specific molecular alterations, rather than prioritizing the tumor origin. Consequently, patients with rare and ultra-rare tumor types, as well as patients with carcinoma of unknown primary (CUP) or patients with multiple prior lines of therapy, have been eligible for targeted therapy, with improved clinical outcomes [14,22,24,25,26,27].

Table 3 summarizes the data from key clinical trials for *NTRK* and *RET* inhibitors, showcasing the benefit of comprehensive genome testing for drug development in the pan-tumor setting.

#### 2.2.1. Neurotrophic Tyrosine Receptor Kinase (NTRK) Fusion-Positive

The three NTRK genes (NTRK1, 2, and 3) encode the tropomyosin receptor kinase (TRK) receptor family proteins responsible for different neural cell functions. Mutations in the NTRK genes (fusions most commonly incriminated) have been observed in solid tumors such as gastrointestinal cancers and gynecological, thyroid, lung, and pediatric malignancies [31,42,43,44].

The inaugural pan-TRK inhibitor Larotrectinib was approved following three pivotal clinical trials, LOXO-TRK-14001, SCOUT, and NAVIGATE, collectively enrolling 153 eligible patients, both adult and pediatric. Pooled analysis showed an Overall Response Rate (ORR) of 79% (95% CI, 72–85%), 16% showing Complete Response (CR), a median duration of response (DoR) of 35.2 months (59 of which had a DoR of >24 months), a median progression-free survival (mPFS) of 28.3 months, and median overall survival (mOS) of 44.4 months. [28,29,45].

Although Entrectinib is also a first-generation, small-molecule inhibitor of NTRK, it is also a multikinase inhibitor of the ROS1 and ALK oncogenes. Approval in the tissue-agnostic setting stems from integrating the data from the ALKA-372-001, STARTRK-1, and STARTRK-2 trials, showing an ORR of 57% (95% CI, 43.2–70.8%), with four patients out of the 54 adult patients included in the trials showing CR. The median DoR (mDoR) was 10 months (95% CI, 7.1 months to not reached), with 56% of patients having a response of over a year. [32,46]. An updated integrated analysis of 121 patients with 14 tumor types and over 30 histologies indicated (after a follow-up period of 25.8 months) an ORR of 61.2%, mDoR of 20 months (95% CI, 13–38.2 months), mPFS of 13,8 months (95% CI, 10.1–19.9 months), and mOS of 33.8 months (95% CI, 23.4–46.4 months) [47].

Repotrectinib is a novel-generation multitargeted tyrosine kinase inhibitor (TKI), with data available for ROS1-positive Non-Small Cell Lung Cancer (NSCLC) through the ongoing TRIDENT-1 trial. [34]. However, the inhibitor showcases activity for NTRK 1/2/3 fusions and is currently on trial in the tumor-agnostic setting (Table 2).

#### 2.2.2. Rearranged During Transfection (RET) Fusion-Positive Cancers

The *RET* gene is located on chromosome 10 and encodes a transmembrane receptor tyrosine kinase that is involved in the normal embryogenesis of the kidney, the enteric nervous system, and spermatogenesis [48,49,50,51,52]. Since the late 1980s, the proto-oncogenic role of *RET* has been identified and characterized in numerous cancer types, including thyroid, lung, and breast cancer, with *RET* alterations occurring in less than 5% of all cancer patients [53,54,55].

Previously, *RET* fusions were targeted using multikinase inhibitors (e.g., Vandetanib). However, the development of selective RET inhibitors, such as Selpercatinib and Pralsetinib, has been advantageous in the pan-cancer treatment setting and received FDA approval due to improved specificity and potency [22].

Selpercatinib is a small-molecule TKI targeting RET alterations (including mutations and fusions) [56] that was FDA-approved after the results from the LIBRETTO-001 trial, which included patients with *RET*-altered tumors, and showed encouraging activity in NSCLC and medullary thyroid carcinoma, as well as other cancers (among 45 patients with non-NSCLC or thyroid carcinoma the ORR was 43.9%, 95% CI 28.5–60.3%) [35,36,37]. In addition to Selpercatinib, Pralsetinib has also shown pan-cancer efficacy across *RET* fusion-positive solid tumors, with an ORR of 57% (95% CI, 35–77%) [38,40]. In the ARROW clinical trial. Further information regarding the LIBRETTO-001 and ARROW clinical trials regarding the specific treatment settings where Selpercatinib and Pralsetinib have been investigated are presented in Table 2.

#### 2.2.3. Von Hippel–Lindau Disease

Von Hippel–Lindau (VHL) disease is a rare hereditary disorder (approximately one in every 27,300–39,000 live births) linked with the development of both benign and malignant neoplasms, including clear-cell renal carcinoma (RCC), pancreatic neuroendocrine tumors (pNETs), and CNS and retinal hemangioblastomas. The hypoxia-inducible factor inhibitor Belzutifan has been approved for the treatment of advanced tumors associated with the germline pathogenic variants of the VHL gene as a result of the LITESPARK-004 trial, which showed promising results, such as an ORR of 59% for RCC (including two CRs), and 90% for pNET, with an acceptable toxicity profile [57,58,59,60,61].

#### 2.2.4. Human Epidermal Growth Factor Receptor 2-Positive (Her2-Positive) Tumors

*HER2* (also known as ERBB2 or Her2-neu) is a 185 kDa transmembrane protein belonging to the Epidermal Growth Factor Receptors family. It is encoded by the *HER2* gene, which is situated on the 17q21 chromosome [62,63,64]. Overexpression of the Her2-neu gene leads to a 40–100-fold increase in HER2 protein, which in turn leads to overexpression on the cellular surface [65]. Although the Her2 is an orphan receptor, the protein relies upon the binding of the extracellular domain of one of the other HER family receptors (HER 1, 3, 4 tyrosine kinases) with one of the 11 possible ligands to undergo heterodimerization (or even homodimerize when expressed at very high levels) and transphosphorylation of the intracellular domain. The phosphorylated products interact with multiple intracellular signaling pathways (such as the phosphoinositide-3-kinase/protein-kinase-B, PI3K/AKT, and anti-apoptosis pathway, for which HER2 is the most potent stimulator), regulating genes involved in cancer cell proliferation, survival, differentiation, angiogenesis, invasion, and metastasis [62,66,67,68,69,70,71]. Although *HER2 overexpression* is classically defined through immunohistochemistry (an IHC3+ expression meaning positivity), the *HER2 amplification* can be interrogated in multigene assays and can be observed in a multitude of solid tumors, including breast, gastric, biliary tract, pancreatic, and lung tumors. As such, NGS can become a practical solution for a larger-scale adoption of anti-*HER2* targeted therapy in the tumor-agnostic setting or after multiple lines of therapy have been exhausted [14,22,72,73].

The DESTINY-PanTumor-02 trial confirmed the pan-cancer efficacy of trastuzumab deruxtecan (T-Dxd) by treating 267 patients across different tumor cohorts (including endometrial, cervical, biliary tract, and pancreatic) with *HER2 overexpression* confirmed through IHC (IHC2+ or IHC3+) with the antibody-drug conjugate. The ORR result was 37.1% for all comers (all patients with centrally confirmed *HER2 overexpression* experienced a response to T-Dxd). However, the response was greater for patients with centrally confirmed IHC3+, with an ORR of 61.3% (95% CI, 49.4–72.4%). Across all cohorts, the mOS was 13.4 months (95% CI, 11.9–15.5 months), ranging from 5 months in the pancreatic cohort to 26 months in the endometrial cohorts. The T-Dxd therapy is not without risks; 40.8% of patients experienced Grade 3 or greater drug-related adverse events, and 10.5% experienced drug-related interstitial lung disease, which in three cases resulted in death [22,72]. Nonetheless, T-Dxd has received approval for the treatment of HER2-positive (IHC3+) solid tumors and is endorsed by guidelines for the treatment of *HER2* overexpressing tumors [14].

Similarly, the MYPathway HER2 basket trial supports HER2-directed therapy with the combination of Pertuzumab and Trastuzumab in HER2-positive (*HER2* overexpressing, IHC3+, or *HER2* amplified) solid tumors, achieving an ORR of 25.9% (notably, patients with IHC3+ expression exhibited an ORR of 41%, patients with IHC2+ expression had an ORR of 21.9%, and no response was observed for patients with no HER2 expression) [73].

#### 2.2.5. BRAF V600E-Mutated Cancers

The aberrant stimulation of the mitogen-activated protein kinase (MAPK) pathway through alterations of BRAF (specifically the substitution of valine with glutamic acid at position 600 of the protein, the BRAF V600E mutation) leads to oncogenesis in up to 3% of people diagnosed with cancer, mainly in thyroid cancer and melanoma. Furthermore, activation of MEK promotes tumor cell growth, proliferation, and survival, and, as such, dual inhibition of BRAF and MEK has been proposed as a therapeutic strategy in BRAF V600E-altered tumors [22,27,74,75,76,77,78,79,80,81,82,83].

After encouraging results from melanoma and NSCLC, the pan-cancer efficacy of Dabrafenib (BRAF inhibitor) and Trametinib (MEK inhibitor) has been evaluated in the Rare Oncology Agnostic Research (ROAR) trial, which indicated clinically meaningful responses in anaplastic thyroid carcinoma (ORR, 56%), biliary tract cancer (ORR, 47%), and gliomas (ORR, 31% for high grade, 69% for low grade), leading toward the approval and recommendation of the dual blockade in the pan-cancer BRAF V600E mutation-positive setting [14,28,84,85,86,87,88].

#### 2.2.6. High Mutational Burden Tumors

The quantitative measure of mutations in the tumor genome (measured as the number of mutations per megabase), as assessed by NGS, is currently under research as a novel positive predictive biomarker for immune checkpoint inhibitors, especially Pembrolizumab. The interest in assessing mutational burden is high, as estimations show that nearly 20% of all cancers have a mutational burden of ≥10 mut/Mb (TMB-H tumors), including melanoma and nonmelanoma skin cancer, and also bladder, lung, and small intestinal cancers [27,89,90,91,92]. As such, compared to the other pan-cancer therapies discussed so far, NGS can be used as a quantitative assay to determine the predictive biomarker for immunotherapy rather than targeted therapy.

Following the KEYNOTE-158 trial, in 102 patients with tumor mutational burden TMB ≥ 10 mut/Mb, treatment with Pembrolizumab resulted in an ORR of 29% (95% CI, 21–39%), with mDoR not reached (half of the patients achieved a duration of response of over two years), mPFS of 2.1 months (95% CI, 2.1–4.1 months), and mOS of 11.7 months (95% CI, 9.1–19.1 months). A higher tumor mutational burden (i.e., ≥13 mut/Mb) correlates with higher response rates, with similar clinical benefits for patients, leading to the FDA approval of Pembrolizumab at the 10 mut/Mb threshold [22,91,93].

#### 2.2.7. Mismatch Repair Deficient (dMMR)/High Microsatellite Instability (MSI-H) Cancers

Similar to TMB-H tumors, dMMR/MSI-H cancers are associated with a hypermutable status, resulting from deficiencies in the mismatch repair genes leading to losses of genomic integrity. These alterations are linked with Lynch Syndrome and are frequently reported in endometrial, small bowel, colon, and gastric cancers [27,94]. Two PD-1 blockers, Pembrolizumab and Dostarlimab, have been FDA-approved for the tissue-agnostic setting, showing improved clinical outcomes [93,95,96,97,98,99,100].

IHC typically determines dMMR/MSI-H status. However, NGS testing can detect mutations in the four key MMR genes (MLH1, PMS2, MSH2, and MSH6) and diagnose Lynch Syndrome, which is an autosomal dominant genetic disease. Thus, NGS can screen and monitor healthy first-degree relatives at risk for Hereditary Non-Polyposis Colorectal Cancer, Endometrial Carcinoma, and other associated cancers. Furthermore, NGS testing can offer a broader panel of actionable biomarkers, impacting further treatment.

Recently, PD-1 antagonists have shown impressive responses in the neoadjuvant setting [101]. Such promising results are desperately awaited in the treatment of locally advanced cancers, especially chemo-unresponsive tumors such as dMMR/MSI-H, where the prognosis remains poor despite multimodal efforts to find a cure [102]. Table 4 summarizes the prospective trials showing the added impact of immune therapy in the pre-surgical setting of dMMR/MSI-H tumors, including the possibility of having a watch-and-wait strategy for tumors showing a complete clinical response after the completion of neoadjuvant immunotherapy.

### 2.3. The Oncogenic Driver Landscape in NSCLC

Determining potential oncogenic drivers is a pivotal step in the diagnosis of NSCLC, as they are positive predictors for targeted therapy, and some are negative predictors for immune checkpoint inhibitors or chemotherapy. The modest response to immune therapy comes as an inherent feature of oncogene-driven tumors, which are characterized by low TMB, lack of neoantigens, and a microenvironment low on immune cells [107]. These key principles of therapy apply to approximately 60% (80% in the Asian population) of lung adenocarcinoma patients who are positive for an oncogenic alteration (Figure 1 presents the incidence for each oncogenic driver). As such, the comprehensive genomic evaluation offered by NGS is vital in the diagnostic work-up for locally advanced and metastatic NSCLC by identifying druggable alterations and screening for de novo or acquired resistance mechanisms [108].

One of the most well-known examples is Epidermal Growth Factor Receptor (EGFR) mutations, present in 10–25% of NSCLC cases, more predominantly in adenocarcinomas, particularly among non-smokers, and more frequent in the Asian population [109]. A total of 90% of EGFR mutations are constituted by the deletion in exon 19 and the leucine–arginine substitution at codon 858 (L858R) and can be targeted by first- (Erlotinib, Gefitinib) and second-generation (Afatinib, Dacomitinib) EGFR inhibitors [110,111]. Unfortunately, the PFS of these drugs is largely dictated by the acquisition of a secondary point mutation, the substitution of methionine for threonine at amino acid position 790 (T790M mutation), conferring tumoral resistance to treatment [112]. The third-generation inhibitor Osimertinib also acts on this mutation, bypassing the resistance mechanism, and is now a pillar in the treatment of EGFR-mutated NSCLC as a first-line metastatic treatment [113,114]. Nonetheless, Osimertinib is also recommended as adjuvant and consolidation therapy after surgery or definitive chemoradiation, showcasing the importance of genomic profiling in pre-metastatic stages of the disease [115,116].

Similar to EGFR mutations, Anaplastic Lymphoma Kinase (ALK) (typically translocations) typically occurs in non-smoker NSCLC patients with adenocarcinoma histotype in approximately 3–5% of NSCLC cases [117,118]. Similarly, the drug development of ALK inhibitors had to overcome acquired resistance mechanisms such as L1196M (which confers resistance to Crizotinib) or ALK G120R/del (which confers resistance to both first- and second-generation ALK inhibitors), leading to the development of Lorlatinib [111,119]. ALK inhibition can also be used in the adjuvant setting, as shown in the ALINA trial, which showed overwhelming benefits for postsurgical treatment with Alectinib [120].

EGFR and ALK mutations are two classic examples of oncogene drivers that must be identified and specifically targeted to substantially benefit patients’ clinical outcomes. As more oncogenes are discovered and therapies are developed, the need for a sensitive investigation to confirm the eligibility to specific targeted therapy tailored for these alterations will become more imperative. Consequently, NGS will become essential, both as a tool for initial diagnosis and indispensable in the later stages of the disease for profiling tumor alterations and correctly choosing treatment by considering druggable mutations and acquired resistance mechanisms. It is thus vital to encourage the development of Molecular Tumor Boards to coordinate treatments, deliver precise therapies, and mitigate the economic burden associated with NGS and novel therapies [121]. Figure 1 and Table 5 provide an overview of the targeted therapy landscape for NSCLC.

### 2.4. Investigating Homologous Repair Deficiencies—Treatment Avenues and Hereditary Cancer Risk Evaluation

Homologous recombination deficiency (HRD) describes the cellular incapacity to repair DNA damage provoked by endogenous or exogenous cancerogenic agents. This results in the accumulation of double-stranded breaks in the DNA helix, increasing the likelihood of developing cancer. HRD can result from germline (as observed in hereditary cancers) or somatic mutations of a multitude of genes, BRCA1 and BRCA2 being the most well-known, due to the increased likelihood of developing breast (56–65% for BRCA1, 35–57% for BRCA2) or ovarian (20–50% for BRCA1, 5–23% for BRCA2) cancers, but also pancreatic, prostate, or colorectal cancers, and with an increased likelihood of synchronous cancers [175,176,177,178]. Other genes have also been linked to HRD, such as ARID1A, ATM, ATRX, BAP1, BARD1, BLM, CHEK1/2, MRE11A, NBN, PALB2, RAD50, RAD51, and WRN, with varying prognostic significance, indicating that evaluating HRD requires interrogation for multiple genetic anomalies simultaneously [178,179,180].

HRD-positivity implies a positive prediction toward treatment with poly (ADP-ribose) polymerase (PARP) inhibitors (PARPi), which have markedly improved outcomes in both adjuvant and metastatic settings [181] (Table 5). Furthermore, HRD positivity has implications for the patient’s family members, as germline mutations cause hereditary cancers in first-degree relatives, requiring inclusion in screening programs and/or prophylactic interventions such as mastectomy or bilateral oophorectomy [181,182]. These interventions need to be tailored according to the cancer penetrance of the alteration detected, and multiple studies have noted the importance of multi-gene panel tests for the detection of hereditary cancers [183,184].

A summary of the pivotal trials regarding PARPi is presented in Table 5. Notably, the inhibitors show activity in BRCA1/2 mutations and the other alterations associated with HRD, showcasing the benefit of a broader investigation. The clinical benefits of improved DFS, PFS, and OS in either adjuvant or metastatic setting cannot be overlooked, as BRCA1/2 mutations are typically associated with poorer prognosis in breast, ovarian, and prostate cancers [185,186,187]. As such, treatment with PARPi offers a much-needed treatment solution for a group of high-risk patients and NGS enables the precise inclusion of the target population, as well as investigating the familial risks for developing cancer (Table 6).

### 2.5. Bridging the Hormone–Chemotherapy Gap in HR-Positive Advanced Breast Cancer

After the development of cyclin-dependent kinase 4 and 6 inhibitors (CDK4/6i), one of the unmet needs in the treatment of hormone receptor-positive (HR+) advanced breast cancer (ABC) was to find therapies that could provide meaningful benefit beyond progression due to the relative chemoresistance of these tumors. Although there have been investigations within the immune-mediated mechanisms of endocrine resistance, so far, immunotherapy is still under investigation and controversial [213,214]. As such, targeted therapies play an important role, particularly in the PIK3/AKT pathway, with PIK3CA mutations prevalent in HR+ breast cancer (34.5%) and AKT1 and PTEN mutations restricted to this subgroup [215]. Furthermore, the ESR1 mutation is of particular interest, as a definitive result yields the certainty of endocrine resistance, and NGS provides a very high diagnostic accuracy [216].

The results of the pivotal trials on targeting this pathway are summarized in Table 7. It is important to note that these alterations should be investigated together with BRCA1/2 mutations (see Section 2.3) for treatment with PARPi and other tumor-agnostic therapies (see Section 2.2.), resulting in an increased need for multigene examination in this subset of patients.

## 3. Discussion

The need for significant developments in the precision oncology era requires tools such as NGS for the therapeutic management of oncologic patients. Several reports have provided evidence for the high-accuracy diagnostic value of NGS [216,222].

It is important to note, however, that several pitfalls may alter the diagnostic accuracy of NGS, particularly in the pre-analytical phase. As such, the tissue blocks selected for analysis must represent a substantial portion of the tumor (at least 20%) for biomolecular analysis viability [223]. Furthermore, factors such as time to fixation, duration of fixation, and the conditions for tissue storage will significantly affect the nucleic acid integrity of the tumor tissue [224]. Nonetheless, sequencing provides challenges as well, with risks including degradation by RNases, cross-linking, and fragmentation [225,226]. These pitfalls need to be considered in the evaluation of the NGS workflow, but disadvantages such as long turnaround times, the need for specialized personnel, and higher costs are outweighed by the pivotal contribution that NGS brings to oncological management.

As the article written by Zalis et al. remarks, coupling NGS testing with liquid biopsies offers significant advantages, such as an alternative to surgical tissue biopsies in a non-invasive manner (reducing the patient discomfort and risk of complications), monitoring cancer progression and response to treatment in real time (allowing for timely adjustments in therapy), detecting minimal residual disease, and identifying resistance mechanisms, thus guiding the selection of alternative treatments [222].

Furthermore, growing evidence suggests that targeted therapy improves response rates and clinical outcomes [227]. A comprehensive literature review carried out by Gibbs et al. indicates a significant impact on survival across tumor types for NGS-informed treatment [227].

Admittedly, NGS and the rise of targeted therapy indications might prove expensive. However, there is growing evidence showing the potential for cost-effectiveness under specific reimbursement policies that would permit adequate access to novel therapies [228]. Furthermore, most targeted therapy options are safer to administer than ChT, with fewer side effects, and require fewer days of hospitalization, as most treatments can be administered orally. This can prove advantageous for the patient’s compliance with treatment, even in adverse situations and unforeseen circumstances, such as COVID-19, which can limit access to hospitals [229].

Compelling economic arguments can also include the dramatic decrease in NGS costs since the early 2000s. Technological advancements and scale economics decreased from $100 million to less than $1000 in some cases. Unfortunately, costs can still prove prohibitive in low- and middle-income countries, especially if a lack of reimbursement means out-of-pocket costs for the patient [230]. This is not to say, however, that NGS analysis and subsequent treatment decisions are not cost-effective, as some systematic reviews have demonstrated financial advantages for NGS-guided treatment decisions [227,231]. Consequently, efforts to develop cost-minimizing workflows for testing, initiatives to develop genomic research capacity, collaborative efforts for genomic data sharing and analysis, and implementation of guidelines mindful of the potential financial toxicity incurred would provide great benefit in the large-scale adoption of NGS testing [232,233].

## 4. Conclusions

In conclusion, NGS has become synonymous with precision oncology, guiding treatments with unparalleled precision and assisting in developing new treatment strategies. Integrating artificial intelligence, collaboration in Molecular Tumor Boards, and continued policy support will be essential for harnessing its full potential and ensuring equitable access to genomic-driven therapy for all patients. Guidelines from the ESMO, NCCN, and ASCO all recommend multigene panel-based genomic testing, and the continuous effort to integrate NGS testing is pivotal for precision oncology.

## 5. Future Directions

Our narrative review showcases the benefits of NGS in guiding diagnosis, targeted therapy, and immune therapy by interrogating multiple complex genomic alterations. The development of new therapies, such as CAR-T cell therapy for solid tumors, may result in new opportunities for NGS adoption. Trials such as the SPEARHEAD-1 in synovial sarcomas show promising results for the adoption of CAR-T cell therapies such as afamitresgene autoleucel and require molecular profiling for melanoma-associated antigen A4 (MAGE-A4) expression and HLA typing [234]. Further trials are expected for such therapies in solid tumors, and the pivotal role of NGS in finding valid targets and HLA typing can play a significant role in this expansion. Furthermore, the considerable impact of NGS in interrogating for polymorphisms of the same gene can be extremely useful in identifying genetic variations of pharmacogenes, implying a near future where therapy toxicity can also be adjusted according to the individual characteristics of each patient [235]. It is, therefore, highly likely that NGS will become synonymous with the practice of precision oncology in the management of solid tumors.

## Figures and Tables

**Figure 1 ijms-26-03123-f001:**
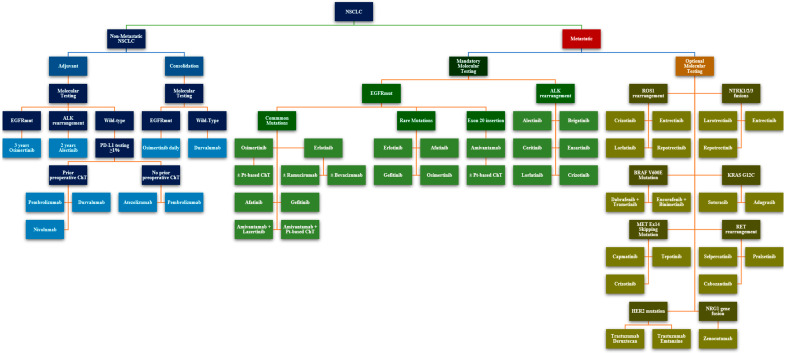
A schematic overview of the treatment options in different clinical scenarios, according to the molecular profile of NSCLC tumors. Indications for targeted therapy were collected from the National Comprehensive Cancer Network (NCCN) Clinical Practice Guidelines in Oncology, Version 4.2025, Published 14th of January 2025. Accessed 24 March 2025 (Available at: https://www.nccn.org/professionals/physician_gls/pdf/nscl.pdf).

**Table 1 ijms-26-03123-t001:** Commercially available NGS assays in the treatment of solid tumors.

NGS Test	Technology Used	Sample Type	Key Notes	Time	Molecular Alterations	Relevant Literature
FoundationOne CDx	Hybrid Capture NGS	Tumor Tissue	Comprehensive solid tumor profiling includes >300 genes and detects fusions and amplifications.	14 days	EGFR, ALK, BRAF, BRCA1/2, MET, NTRK, ROS1, PIK3CA, RET, ERBB2 (HER2), KRAS, IDH1, FGFR1/2/3, CDK4, CDK6, KIT, PDGFRA, TSC1/2, ERBB3	[5,6]
FoundationOne Liquid CDx	cfDNA Sequencing	Blood (Liquid Biopsy)	Liquid biopsy test; sensitivity decreases for variants with allele frequency <0.1%.	10–14 days	EGFR, ALK, BRAF, BRCA1/2, MET, NTRK, ROS1, PIK3CA, RET, ERBB2 (HER2), KRAS, IDH1, FGFR2/3	[7]
Guardant360 CDx	cfDNA Sequencing	Blood (Liquid Biopsy)	High sensitivity for detecting rare cfDNA variants; optimized for minimal input samples.	7 days	EGFR, ALK, BRAF, BRCA1/2, MET, RET, ERBB2 (HER2), PIK3CA, KRAS, IDH1	[8]
MSK-IMPACT	Hybrid Capture NGS	Tumor Tissue	Highly validated for solid tumors; includes 468 cancer-relevant genes.	21 days	EGFR, ALK, BRAF, BRCA1/2, MET, NTRK, ROS1, PIK3CA, RET, ERBB2 (HER2), KRAS, IDH1, FGFR1/2/3, CDK4, CDK6, KIT, PDGFRA, TSC1/2, ERBB3	[9]
Oncomine Dx Target Test	AmpliSeq NGS	Tumor Tissue	Focused on NSCLC; detects EGFR, ALK, ROS1, BRAF, and other actionable mutations.	7–10 days	EGFR, ALK, BRAF, ROS1, RET, MET, KRAS	[10]
Oncomine Comprehensive Assay	AmpliSeq NGS	Tumor Tissue	Comprehensive tumor profiling with >500 genes, including fusion detection.	7–14 days	EGFR, ALK, BRAF, ROS1, RET, MET, KRAS	[11]
Tempus xT	Hybrid Capture NGS	Tumor Tissue	The broad panel covering >600 genes; includes TMB and MSI analysis.	10–14 days	EGFR, ALK, BRAF, BRCA1/2, MET, NTRK, ROS1, PIK3CA, RET, ERBB2 (HER2), KRAS, IDH1, FGFR1/2/3, CDK4, CDK6, KIT, PDGFRA, TSC1/2, ERBB3	[12]
Caris Molecular Intelligence	Multiple NGS Platforms	Tumor Tissue	Uses multiple platforms; integrates NGS, IHC, and other molecular tests for a holistic profile.	10–14 days	EGFR, ALK, BRAF, BRCA1/2, MET, NTRK, ROS1, PIK3CA, RET, ERBB2 (HER2), KRAS, IDH1, FGFR1/2/3, CDK4, CDK6, KIT, PDGFRA, TSC1/2, ERBB3	[13]
Com.Pl.it DX Colon	Hybrid Capture NGS	Tumor Tissue	Focused on colorectal cancer profiling; including MSI detection and fusion analysis.	10–14 days	EGFR, ALK, BRAF, KRAS, MET, RET, PIK3CA, ERBB2 (HER2), MSI detection	
Com.Pl.it DX Liquid Biopsy	cfDNA Sequencing	Blood (Liquid Biopsy)	Designed for liquid biopsy; slightly lower sensitivity than tissue-based assays for rare variants.	10–14 days	EGFR, ALK, BRAF, KRAS, MET, RET, PIK3CA, ERBB2 (HER2), FGFR1/2/3, MSI	

List of abbreviations: ALK—Anaplastic Lymphoma Kinase. BRAF—B-Raf Proto-Oncogene. BRCA1/2—Breast Cancer Gene 1/2. CDK4/CDK6—Cyclin-Dependent Kinases 4 and 6. cfDNA—Circulating Free DNA. EGFR—Epidermal Growth Factor Receptor. ERBB2 (HER2)—Erb-B2 Receptor Tyrosine Kinase 2 (also known as HER2). ERBB3—Erb-B2 Receptor Tyrosine Kinase 3. FGFR1/2/3—Fibroblast Growth Factor Receptor 1, 2, and 3. IDH1—Isocitrate Dehydrogenase 1. IHC—Immunohistochemistry. KIT—KIT Proto-Oncogene, Receptor Tyrosine Kinase. KRAS—Kirsten Rat Sarcoma Viral Oncogene. MET—MET Proto-Oncogene. MSI—Microsatellite Instability. NGS—Next-Generation Sequencing. NTRK—Neurotrophic Tyrosine Receptor Kinase. NSCLC—Non-Small Cell Lung Cancer. PDGFRA—Platelet-Derived Growth Factor Receptor Alpha. PIK3CA—Phosphatidylinositol-4,5-Bisphosphate 3-Kinase Catalytic Subunit Alpha. RET—Rearranged During Transfection. ROS1—ROS Proto-Oncogene 1. TMB—Tumor Mutational Burden. TSC1/2—Tuberous Sclerosis Complex 1 and 2.

**Table 2 ijms-26-03123-t002:** Molecular markers for the diagnosis and management of gliomas, as recommended by the 2021 EANO Guidelines.

Molecular Marker	Biological Function of Affected Genes	Diagnostic Role	Therapeutic Value
IDH1 (R132) or IDH2 (R172) mutation	Gain-of-function mutation	Distinguishes diffuse gliomas with IDH mutation from IDH-wild-type glioblastomas or other gliomas	Predictive biomarker for treatment with Vorasidenib in IDH-1 or IDH-2-mutant low-grade gliomas
1p/19q codeletion	Inactivation of putative tumor suppressor genes on 1p (such as FUBP1) and 19q (such as CIC)	Distinguishes IDH-mutant oligodendroglioma and 1p/19q-codeleted from IDH-mutant astrocytoma	Different therapeutic management
Loss of nuclear ATRX	Cell proliferation and longevity promoter with telomere lengthening activity	Characteristic of tumors of astrocytic lineage	Specific therapeutic management of astrocytic tumors
Histone H3 K27M	Missense mutations affect the epigenetic regulation of gene expression	Characteristic of the particular diffuse midline glioma, H3 K27M-mutant	Specific therapeutic management, radiotherapy alone as adjuvant treatment permitted
Histone H3.3 G34R/V mutation	Mutation that affects the epigenetic regulation of gene expression	Characteristic of the particular diffuse hemispheric glioma, H3.3 G34-mutant	Specific adjuvant management with Temozolomide chemoradiotherapy as standard
MGMT promoter methylation	DNA repair	None	Predictive biomarker for the benefit of alkylating ChT agents in IDH-wt glioblastoma (very important in frail or elderly patients)
Homozygous deletion of CDKN2A/CDKN2B	Encode kinase inhibitors and tumor suppressors functioning as regulators of Rb1 and p53-dependent signaling	Marker of poor outcome and WHO Grade 4 disease in IDH-mutant astrocytoma.	Specific adjuvant management with Temozolomide chemoradiotherapy
EGFR amplification	Impact on cellular proliferation, invasion, and resistance to induction of apoptosis	Molecular marker characteristic of glioblastoma, IDH wild type, WHO Grade 4	Specific therapeutic management–Stupp protocol, with adjuvant Temozolomide for 6 months following Temozolomide chemoradiotherapy
TERT promotor alteration	Cell proliferation marker, which promotes cellular longevity via increased TERT expression	Molecular marker of glioblastoma, IDH wild type, WHO Grade 4	Specific therapeutic management–Stupp protocol, with adjuvant Temozolomide for 6 months following Temozolomide chemoradiotherapy
+7/−10 cytogenetic signature	Gain of chromosome 7 combined with loss of chromosome 10, meaning gain of genes such as PDGFA and EGFR simultaneous with the loss of PTEN and MGMT	Molecular marker of glioblastoma, IDH wild type, WHO Grade 4	Specific therapeutic management–Stupp protocol, with adjuvant Temozolomide for 6 months following Temozolomide chemoradiotherapy
BRAF V600E mutation	Oncogenic driver mutation that leads to the activation of the MAPK pathway	Rare in adult diffuse gliomas	Amenable to pharmacological intervention with BRAF inhibitors (Dabrafenib ± Trametinib, Vemurafenib ± Cobimetinib)

**Table 3 ijms-26-03123-t003:** Drug development in the tumor-agnostic setting—available data from key clinical trials for *NTRK* and *RET* inhibitors.

Therapy	Targeted Molecular Alteration(s)	ESCAT	Key Clinical Trial(s) and Relevant Literature	Molecular Tests Performed	Tumors Included	Trial Design	Trial Outcomes	Toxicity and AEs From
Larotrectinib (First-Generation NTRKi)	TRKA/B/C	IC	NAVIGATE [28] (NCT02576431 NCT02122913)	NGS, FISH	Soft tissue sarcomas, salivary gland tumors, thyroid, lung, colorectal, melanoma, breast, pancreatic, and primary CNS tumors.	Phase 2 basket trial that enrolled 55 patients, both adults and children, with TRK fusion-positive tumors.		The most common are fatigue, dizziness, and nausea. Serious (Grade ≥ 3): 13% of patients, primarily elevated liver enzymes. No treatment-related deaths.
			SCOUT [29] (NCT02637687)	NGS, FISH	Infantile fibrosarcoma, thyroid, CNS tumors, and other solid tumors in children.	Phase 1/2 trial enrolled 24 pediatric patients with locally advanced or metastatic solid or CNS tumors, 17 of which had NTRK fusion-positive tumors. Larotrectinib was administered orally, twice daily, on a continuous 28 day cycle in increasing dose levels.	Demonstrated antitumor activity for all patients with TRK fusion-positive tumors at a dose of 100 mg/m^2^ (cap of 100 mg), ORR 93%.	
			VICTORIA [30]	NA		Protocol-driven, an exact-matching analysis comparing 82 patients with TRK fusion-positive cancers treated with Larotrectinib in clinical trials with TRK fusion-positive treated with other non-TRK inhibitor therapies in a real-world setting.	Improved OS (NR vs. 37.2 months, HR 0.44, 95% CI 0.23–0.83), longer time to next therapy, duration of therapy, and PFS (36.8 vs. 5.2 months, HR 0.29, 95% CI 0.18–0.46).	
Entrectinib (First-Generation NTRKi)	TRKA/B/C, ROS1, ALK	IC	ALKA-372-001 (EudraCT 2012-000148-88) and STARTRK-1 (NCT02097810) [31]	NGS, FISH	NSCLC, colorectal, breast, pancreatic, thyroid, and salivary gland tumors.	Two Phase I trials assessed the safety, tolerability, and preliminary efficacy in 119 adults with advanced or metastatic tumors harboring fusions in the NTRK1/2/3, ROS1, or ALK genes.	Demonstrated a favorable safety profile with preliminary antitumor activity and established the recommended dose of 600 mg of Entrectinib daily.	Most common: fatigue, dysgeusia, dizziness. Serious (Grade ≥ 3): eosinophilic myocarditis, cognitive disturbance, fatigue—resolved with dose interruption. No treatment-related deaths.
			STARTRK-2 [32] (NCT02568267)	NGS, FISH	NSCLC, colorectal, breast, pancreatic, thyroid, soft tissue sarcomas, and salivary gland tumors.	Phase II, basket trial, evaluating the efficacy and safety of Entrectinib 600 mg daily in patients with tumors with NTRK1/2/3, ROS1, and ALK gene functions.	ORR for NTRK fusion-positive tumors: 57.4% (95% CI 43.2–70.8); mDoR 10.4 months; mPFS 11.2 months; mOS 20.9 months.	Most common: Grade 1–2 dysgeusia (47.1%), constipation (27.9%), fatigue (27.9%). Serious (Grade ≥ 3): anemia (11.8%), weight increase (10.3%). No treatment-related deaths.
			STARTRK-NG [33] (NCT02650401)	NGS, FISH	Neuroblastoma, infantile fibrosarcoma, glioblastoma, high-grade gliomas, medulloblastoma, soft tissue sarcomas, Ewing sarcoma, rhabdomyosarcoma.	Phase I/II, open-label, dose-escalation, and expansion study in 47 pediatric and young adult patients with solid tumors or primary CNS tumors harboring NTRK1/2/3, ROS1, or ALK gene fusions.	Established the recommended phase II dose for pediatric patients; observed objective responses in NTRK fusion-positive tumors (ORR 57.7%, 95% CI 36.9–76.7, mDoR NR, mDoT 10.6 months, 4.2–18.4 months), indicating potential efficacy in the pediatric population.	Safety profile consistent with adult studies; common AEs included fatigue, gastrointestinal symptoms, and increased liver enzymes; no new safety signals were identified in the pediatric population.
Repotrectinib (Second-Generation NTRKi)	TRKA/B/C, ROS1, ALK	IC	(ongoing) TRIDENT-1 [34] (NCT03093116)	NGS	NSCLC, colorectal, breast, pancreatic, thyroid, soft tissue sarcomas, and salivary gland tumors.	Phase I/II, open-label, multicenter study in patients with advanced solid tumors harboring specific gene fusions. Includes both treatment-naïve and pretreated patients.	Current data are available only for ROS1 fusion-positive advanced NSCLC [34]. The trial determined a recommended dose of 160 mg daily Repotrectinib for 14 days, followed by 160 mg twice daily. For treatment-naïve tumors: ORR 79% (95% CI, 68–88%), mDoR 34.1 months, mPFS 35.7 months. For ROS1 TKI-pretreated patients: ORR 38% (95% CI, 25–52%), mDoR 14.8 months, mPFS 9 months. Data are immature for NTRK-positive tumor agnostics.	Most common: dizziness (58%), dysgeusia (50%), paresthesia (30%). Serious (Grade ≥ 3): reported in 29% of patients—anemia, increased blood creatine kinase levels, dizziness. No treatment-related deaths.
Selpercatinib	RET gene fusions and mutations	IC	LIBRETTO-001 [35,36,37] (NCT03157128)	NGS, FISH	NSCLC, MTC, and PTC thyroid cancer, pancreatic cancer, and colorectal cancer.	Ongoing Phase I/II, open-label trial in patients with RET fusion-positive or RET-mutant cancers.	NSCLC (*n* = 105): ORR 85%; mDoR: 18.4 months. Thyroid (MTC, *n* = 143; PTC, *n* = 19): MTC: ORR 69%; mPFS: 23.6 months. PTC: ORR 79%. Agnostic non-thyroid, non-NSCLC, *n* = 45: ORR 57%; mDoR: 13.0 months.	Most common: dry mouth, diarrhea, hypertension, fatigue. Serious (Grade ≥ 3): hypertension, increased liver enzymes. No treatment-related deaths.
Pralsetinib	RET gene fusions and mutations	IC	ARROW [38,39,40,41] (NCT03037385)	NGS, FISH	NSCLC, MTC, PTC, pancreatic cancer, cholangiocarcinoma, neuroendocrine, SCLC, unknown primary, thymic, ovarian, head and neck, colorectal.	Phase I/II, open-label study evaluating RET fusion-positive or RET-mutant cancers.	NSCLC (*n* = 216): ORR 70% (95% CI: 62–77); mPFS: 17.1 months. Thyroid (RET-mutant MTC and RET fusion-positive thyroid cancer, *n* = 29): ORR 60%; mPFS: not reached. Agnostic (non-thyroid, non-NSCLC, *n* = 38): ORR 57% (95% CI: 42–71).	Most common: constipation (30%), dry mouth (26%), hypertension (24%), fatigue (23%). Serious (Grade ≥ 3): hypertension (15%), neutropenia (10%), anemia (8%). No treatment-related deaths.

List of abbreviations: AE: Adverse Events, ALK: Anaplastic Lymphoma Kinase, CI: Confidence Interval, CNS: Central Nervous System, DoR: Duration of Response, ESCAT: European Society for Medical Oncology Scale for Clinical Actionability of Molecular Targets, FISH: Fluorescence In Situ Hybridization, IC: International Classification, mDoR: Median Duration of Response, mDoT: Median Duration of Therapy, mOS: Median Overall Survival, mPFS: Median Progression-Free Survival, MTC: Medullary Thyroid Carcinoma, NA: Not Available, NGS: Next-Generation Sequencing, NR: Not Reached, NSCLC: Non-Small Cell Lung Cancer, ORR: Objective Response Rate, PFS: Progression-Free Survival, PTC: Papillary Thyroid Carcinoma, RET: Rearranged During Transfection, ROS1: c-ros Oncogene 1, SCLC: Small Cell Lung Cancer, TRKA/B/C: Tropomyosin Receptor Kinase A/B/C, TKI: Tyrosine Kinase Inhibitor.

**Table 4 ijms-26-03123-t004:** Prospective clinical trials that are investigating immune checkpoint inhibitors as neoadjuvant therapy for MSI-H gastrointestinal cancers.

Trial Name	Immunotherapy	Indication	Neoadjuvant Regimen	Design	MSI-H (No. of Patients)	Clinical Outcome
NEONIPIGA [103] (NCT04006262)	Nivolumab plus Ipilimumab	Localized esophageal and gastric adenocarcinoma	Nivolumab 240 mg once every 2 weeks ×6 and ipilimumab 1 mg/kg once every 6 weeks ×2, followed by surgery and adjuvant nivolumab 480 mg once every 4 weeks (nine injections).	Phase II, prospective-single-arm, open-label	32	29/32 patients underwent surgery, 17/29 had pCR.
NICHE-1 [104] (NCT03026140)	Nivolumab plus Ipilimumab	Colon cancer	1 mg/kg Ipilimumab (1 dose) and 3 mg/kg Nivolumab (2 doses), followed by surgery.	Phase II, prospective-single-arm, open-label	20	All patients responded, with 19/20 having an MPR and 12/20 achieving pCR.
NICHE-2 [105] (NCT03026140)	Nivolumab plus Ipilimumab	Colon cancer	1 mg/kg Ipilimumab (1 dose) and 3 mg/kg Nivolumab (2 doses), followed by surgery.	Phase II, prospective-single-arm, open-label	115	113/115 had surgery, 109/111 had a response, 105 had MPR, and 75 achieved pCR.
Cercek et al. [101] (NCT04165772)	Dostarlimab	Rectal cancer	500 mg q3w for 6 months (9 cycles), followed by RT (50.4 Gy/28 fractions with concurrent Capecitabine at standard doses) and then TME if no cCR.	Phase II, prospective-single-arm	12	100% cCR
Ludford et al. [106] (NCT04082572)	Pembrolizumab	Colorectal cancer	200 mg q3w for 6 months, followed by surgical resection, with an option to continue therapy for 1 year.	Phase II, open-label, single-center trial	35 (27 CRC, 8 non-CRC)	17/35 underwent surgery. Of these, 14 had CRC, and 11/14 had pCR.

List of abbreviations: CRC: Colorectal Cancer. cCR: Clinical Complete Response. MPR: Major Pathological Response. MSI-H: Microsatellite Instability-High. pCR: Pathological Complete Response. q3w: Every 3 Weeks. RT: Radiotherapy. TME: Total Mesorectal Excision.

**Table 5 ijms-26-03123-t005:** Pivotal trials in oncogenic-driven NSCLC with targeted therapy, trial design, objective response rate (ORR), median progression/disease-free survival in months (mPFS/DFS), with hazard ratio (HR) if the trial was Phase 3, and median overall survival (mOS) in months, with HR if the trial was Phase 3. Indications for targeted therapy were collected from the National Comprehensive Cancer Network (NCCN) Clinical Practice Guidelines in Oncology, Version 3.2025 Published 14th of January 2025, Accessed 24 March 2025 (Available at: https://www.nccn.org/professionals/physician_gls/pdf/nscl.pdf).

Targetable Oncogenic Alteration	Drug	Setting	Trial	Design	ORR	mPFS/DFS in mo. (HR)	mOS in mo. (HR)
EGFR typical and atypical mutations	Afatinib *	First line	LUX-Lung 3/6 [122,123,124] (NCT00949650; NCT01121393)	Phase 3 open-label trial comparing Afatinib to platinum-based ChT in Asian and Western patients.	66%	11.0 vs. 6.9 (HR: 0.58)	27.3 vs. 24.3 (HR: 0.81)
		Subsequent post-ChT	LUX-Lung 2 [125,126] (NCT00525148)	Phase 2 single-arm trial	58%	4.4	24.8
	Erlotinib *	First line	EURTAC [127] (NCT00446225)	Phase 3 open-label trial comparing Erlotinib to ChT	58%	9.7 vs. 5.2 (HR: 0.37)	19.3 (NR for chemo)
		Subsequent post-ChT	NCIC-CTG [128]	Phase 3 open-label trial comparing Erlotinib to ChT	8.9%	2.2 vs. 1.8 (HR: 0.61)	6.7 vs. 4.7 (HR: 0.70)
	Dacomitinib *	First line	ARCHER1050 [129,130] (NCT01774721)	Phase 3 double-blind trial comparing Dacomitinib to Gefitinib	74.9%	14.7 vs. 9.2 (HR: 0.59)	34.1 vs. 26.8 (HR: 0.76)
	Gefitinib *	First line	NEJ002 [131,132] (UMIN-CTR, C000000376)	Phase 3 open-label trial comparing Gefitinib to ChT	73.7%	10.8 vs. 5.4 (HR: 0.30)	30.5 vs. 23.6 (HR: 0.72)
	Osimertinib *	Adjuvant	ADAURA [115,133] (NCT02511106)	Phase 3 double-blind trial comparing Osimertinib to placebo as adjuvant 3 year-treatment after surgical resection of Stage IB-IIIA NSCLC	NA	NR vs. 27.5 months (HR: 0.20)	5-yr OS: 88% vs. 78% (HR: 0.49)
		Post-RT consolidation	LAURA [116] (NCT03521154)	Phase 3 double-blind trial comparing Osimertinib to placebo as consolidation treatment	NA	39.1 vs. 5.6 (HR: 0.16)	Data not yet mature
		First line	FLAURA [113,134] (NCT02296125)	Phase 3 double-blind trial comparing Osimertinib to first-generation EGFR TKI in first-line metastatic EGFR-positive (exon 19 deletion or L858R) NSCLC	80%	18.9 vs. 10.2 (HR: 0.46)	38.6 vs. 31.8 (HR: 0.80)
		First line + ChT	FLAURA2 [114] (NCT04035486)	Ongoing Phase 3 trial investigating the addition of chemotherapy to first-line Osimertinib	83%	29.4 vs. 19.9 (HR 0.62)	Data not yet mature
		Subsequent post-ChT	AURA3 [135,136] (NCT02151981)	Phase 3 double-blind trial comparing Osimertinib to platinum-pemetrexed chemotherapy for treatment of T790M EGFR-mutant metastatic NSCLC.	71%	10.1 vs. 4.4 (HR: 0.30)	26.8 vs. 22.5 (HR: 0.87)
	Amivantamab	First line + ChT	PAPILLON [137] (NCT04538664)	Ongoing Phase 3 trial investigating the addition of Amivantamab to platinum ChT in EGFR exon 20 insertion mutations	73%	11.4 vs. 6.7 (HR: 0.40)	Data not yet mature
		First line + Lazertinib	MARIPOSA [138] (NCT04487080)	Ongoing Phase 3 trial investigating the addition of Amivantamab to Lazertinib, in comparison with Osimertinib	86%	23.7 vs. 16.6 (HR: 0.70)	Data not yet mature
HER2 mutation (IHC3+)	T-DXd	Subsequent lines	DESTINY-Lung02 [139] (NCT04644237)	Phase 2 single-arm trial investigating T-DXd for previously treated (platinum-ChT) patients with Her2 mutation-positive patients	49%	9.9	19.5
KRAS G12C	Sotorasib	Subsequent lines	CodeBreak200 [140] (NCT04303780)	Phase 3 double-blind trial comparing Sotorasib to Docetaxel in patients with previously treated metastatic NSCLC with KRAS G12C mutation	28%	5.6 vs. 4.5 (HR: 0.66)	10.6 vs. 11.3 months (HR: 1.01)
	Adagrasib	Subsequent lines	KRYSTAL-1 [141] (NCT03785249)	Phase 2 single-arm trial	42.9%	6.5	12.6
RET rearrangement	Selpercatinib	First line	LIBRETTO-001 [37] (NCT03157128)	Phase 1/2 basket trial	84%	20.3	Not available
	Pralsetinib	First line	ARROW [40] (NCT03037385)	Phase 1/2 basket trial	70%	16.5	Not available
	Cabozantinib	Subsequent lines	Drilon et al, 2016 [142] NCT01639508	Phase 2, single-center, open-label, Simon two-stage trial investigating the effectiveness of Cabozantinib in RET rearranged metastatic NSCLC	31%	5.5	9.9
ALK rearrangement	Crizotinib	First line	PROFILE 1014 [143] (NCT01154140)	Phase 3 double-blind trial comparing Crizotinib to ChT in the first line setting for ALK fusion-positive metastatic NSCLC	74%	10.9 vs. 7 (HR 0.45)	47.5 vs. 47.7 (HR 1.00)
		Subsequent post-ChT	Shaw et al, 2013 [144] (NCT00932893)	Phase 3 open-label trial comparing Crizotinib to ChT in previously ChT-treated ALK fusion-positive metastatic NSCLC patients	65%	7.7 vs. 3.0 (HR 0.49)	20.3 vs. 22.8 (HR 1.02)
	Ceritinib	First line	ASCEND-4 [145,146] (NCT01828099)	Phase 3 open-label trial comparing Ceritinib to ChT	72.5%	16.6 vs. 8.1 (HR: 0.55)	NR vs. 26.2 (HR 0.73)
		Subsequent post-ChT	ASCEND-5 [147] (NCT01828112)	Phase 3 double-blind trial comparing Ceritinib to ChT	39.1%	5.4 vs. 1.6 (HR: 0.49)	NR vs. 20.0
	Brigatinib	First line	ALTA-1L [148,149,150] (NCT02737501)	Phase 3 open-label trial comparing Brigatinib to Crizotinib	74%	24.0 vs. 11.0 (HR: 0.49)	NR vs. 52.2
		Subsequent lines	ALTA [151,152] (NCT02094573)	Phase 2 single-arm trial investigating Brigatinib in Crizotinib-refractory ALK fusion-positive metastatic NSCLC	53%	15.6	34.1
	Alectinib	First/subsequent lines	ALEX [153,154] (NCT02075840)	Phase 3 open-label trial, compared with Crizotinib	82.9%	25.8 vs. 12.7 (HR: 0.51)	5-yr OS: 62.5% vs. 45.5% (HR 0.67)
	Ensartinib	First line	eXalt3 [155] (NCT02767804)	Phase 3 open-label trial, compared with Crizotinib	75%	25.8 vs. 12.7 (HR: 0.51)	5-yr OS: 78% vs. 78%
	Lorlatinib	First/subsequent lines	CROWN [119,156,157] (NCT03052608)	Phase 3 double-blind trial, compared with Crizotinib	81%	NR vs. 9.1 (HR: 0.19)	Data not yet mature
ROS1 rearrangement	Crizotinib	First line	PROFILE 1001 [158,159] (NCT00585195)	Phase 1/2 single-arm trial	72%	19.2	51.4
	Entrectinib	First line	STARTRK-1, -2 and ALKA-001 [160,161] (NCT02097810; NCT02568267; EudraCT, 2012–000148–88	Integrated analysis of three Phase 1/2 trials investigating the efficiency of Entrectinib in ROS1 rearrangement-positive tumors	77%	19	47.8
	Repotrectinib	First/subsequent lines	TRIDENT-1 [34] (NCT03093116)	Phase 1/2 basket trial, investigating first line or subsequent line Repotrectinib in tumors harboring ROS1, NTRK1-3, ALK gene fusions, including metastatic NSCLC.	79% in the first line	35.7 in the first line	Not estimable in the first line
					38% post-TKI	9 post-TKI	25.1 post-TKI
	Lorlatinib	First/subsequent lines	Shaw et al, 2019 [162] (NCT01970865)	Open-label, single-arm Phase 1/2 trial investigating Lorlatinib in ROS1-positive NSCLC	62% in the first line	21.0 in the first line	Not available
					35% post-TKI	8.5 post-TKI	Not available
MET exon 14 skipping mutation	Capmatinib	First/subsequent lines	GEOMETRY mono-1 [163,164] (NCT02414139)	Phase 2 single-arm trial	41% in first line	12.5 in first line	21.4 in first line
					44% post-TKI	5.5 post-TKI	16.8 in first line
	Tepotinib	First/subsequent lines	VISION [165,166] (NCT02864992)	Phase 2 single-arm trial	57.3% in first line	12.6 in first line	21.3 in first line
					45.0% in pretreated	11 in pretreated	19.3 in pretreated
	Crizotinib	First/subsequent lines	Drilon et al., 2021 [167] (NCT00585195)	Phase 2 single-arm trial	32%	7.3	20.5
BRAF V600E	Dabrafenib + Trametinib	First/subsequent lines	Planchard et al, 2017 [168,169] (NCT01336634)	Phase 2 single-arm	63.9% in first line	10.8 in first line	17.3 in first line
					68.4% in pretreated	10.2 in pretreated	18.2 in pretreated
	Encorafenib + Binimetinib	First/subsequent lines	Riely et al, 2023 [170] (NCT03915951)	Phase 2 single-arm	75% in first line	NE in first line	Not available
					46% in pretreated	9.3 in pretreated	
NTRK1/2/3 gene fusion	Larotrectinib	First/subsequent lines	Drilon et al, 2018 [171] (NCT02576431 NCT02122913)	Retrospective analysis of multiple Phase 1/2 single-arm trials	73%	35.4	40.7
	Entrectinib	First/subsequent lines	Paz-Ares et al, 2019 [172,173] ALKA (EudraCT 2012-000148-88), STARTRK-1, -2 (NCT02097810; NCT02568267)	Integrated analysis of multiple Phase 1/2 single-arm trials enrolling patients treated with Entrectinib for NTRK fusion-positive tumors, including metastatic NSCLC	62.7%	27.3	41.5
	Repotrectinib	First/subsequent lines	TRIDENT-1 (NCT03093116)	Ongoing Phase 1/2 basket trial, results published for patients with ROS1 fusion (see above).	Not Available	Not Available	Not Available
NRG1 gene fusion	Zenocutuzumab	Subsequent lines	eNRGy [174] (NCT02912949)	Ongoing Phase 2 single-arm trial	Not Available	Not Available	Not Available

* List of abbreviations: First line: First-line therapy, BRAF: B-Raf Proto-Oncogene, ChT: Chemotherapy, DFS: Disease-Free Survival, HR: Hazard Ratio, MET: Mesenchymal-Epithelial Transition factor, mOS: Median Overall Survival, mPFS: Median Progression-Free Survival, NRG1: Neuregulin 1, NTRK: Neurotrophic Tyrosine Receptor Kinase, ORR: Objective Response Rate, RET: Rearranged during Transfection, ROS1: Receptor Tyrosine Kinase, Subsequent lines: Treatments given after prior therapies, TKI: Tyrosine Kinase Inhibitor.

**Table 6 ijms-26-03123-t006:** Pivotal Phase 3 trials in PARP inhibitor therapy across cancer types, including trial design, median progression/disease-free survival in months (mPFS/DFS), hazard ratio (HR), and median overall survival (mOS) in months with HR.

PARP inhibitor	Cancer Treated	Clinical Setting	Clinical Trial	Trial Design	mPFS/DFS in mo. (HR)	mOS in mo. (HR)
Olaparib	Ovarian	Maintenance post-ChT	SOLO-1 [188,189] (NCT01844986)	Phase 3 double-blind trial vs. placebo, which included 391 patients with gBRCA1/2m, treated until PD/unacceptable toxicity or no evidence of disease after 2 years of treatment with Olaparib 300 mg bid.	56.0 vs. 13.8 (0.33)	NR vs. 75.2 (0.55)
		Maintenance post-ChT (with Bevacizumab)	PAOLA-1 [190,191] (NCT02477644)	Phase 3 double-blind trial vs. placebo, which included 806 patients with HRD-positive tumors, treated with Bevacizumab 15 mg/kg IV q3w until PD/unacceptable toxicity, or up to 15 months of treatment (including the period when Bevacizumab was offered as primary treatment), and Olaparib 300 mg bid as maintenance until PD/unacceptable toxicity or up to 2 years.	22.1 vs. 16.6 (0.59)	75.2 vs. 57.3 (0.62)
	Breast	Adjuvant	OlympiA [192,193] (NCT02032823)	Phase 3 double-blind trial vs. placebo, which included 1836 patients with gBRCA1/2m high-risk, HER2-negative early breast cancer for 1 year of treatment with adjuvant 300 mg bid Olaparib.	NR (Δ4yr-iDFS = 7.3%, HR: 0.63)	NR (Δ4yr-OS = 3.4%, HR: 0.68)
		Metastatic	OlympiAD [194,195,196] (NCT02000622)	Phase 3 open-label trial vs. chemotherapy (physician’s choice), which included 302 patients with gBRCA1/2m and HER2-negative advanced breast cancer who received Olaparib 300 mg bid until PD/unacceptable toxicities.	7.0 vs. 4.2 (0.58)	22.6 vs. 14.7 (0.55)
	Prostate	Second line mCRPC (monotherapy)	PROfound [197] (NCT02987543)	In a Phase 3 open-label trial, Cohort A included 245 patients with alterations in BRCA1, BRCA2, or ATM, previously treated with Enzalutamide or Abiraterone. Patients were randomized 2:1 to receive Olaparib 300 mg bid or a novel hormonal agent (physician’s choice between Enzalutamide or Abiraterone).	7.4 vs. 3.6 (0.34)	18.5 vs. 15.1 (0.64)
		First line mCRPC (with Abiraterone)	PROpel [198,199] (NCT03732820)	Phase 3 double-blind trial vs. placebo, which included 796 patients with mCRPC who received Olaparib 300 mg bid plus Abiraterone 1000 mg daily vs. placebo plus Abiraterone as first-line treatment, regardless of HRD status.	24.8 vs. 16.6 (HR: 0.66)	42.1 vs. 34.7 (HR: 0.81)
	Pancreatic	Maintenance	POLO [200,201] (NCT02184195)	Phase 3 double-blind trial vs. placebo. Patients with gBRCA1/2m pancreatic adenocarcinoma who did not progress after 16 weeks of first-line platinum-based ChT received maintenance Olaparib 300 mg bid until PD/unacceptable toxicity.	7.4 vs. 3.8 (HR: 0.53)	19.0 vs. 19.2 (HR: 0.83; not significant)
	Endometrial	Advance/Recurrent	DUO-E/GOG-3041/ENGOT-EN10 [202] (NCT04269200)	Phase 3 double-blind trial evaluating Olaparib in combination with Durvalumab as maintenance therapy after primary ChT plus Durvalumab for 718 patients with advanced endometrial cancer.	15.1 vs. 9.6 (HR: 0.55)	NR vs. 25.9 (HR: 0.59)
Niraparib	Ovarian	Maintenance post-ChT	PRIMA [203,204] (NCT02655016)	Phase 3 double-blind trial vs. placebo, investigating maintenance Niraparib 300 mg daily for up to 36 months in patients who responded to primary platinum-based ChT, regardless of HRD status.	13.8 vs. 8.2 (HR: 0.62 in the overall population) 21.9 vs. 10.4 (HR: 0.43 in patients with HRD-positive)	No significant difference in OS in the overall population.
	Prostate	mCRPC (with Abiraterone)	MAGNITUDE [205] (NCT03748641)	Phase 3 double-blind trial vs. placebo, investigating the efficiency of Niraparib added to Abiraterone in 423 patients with HRD-positive and 247 with no HRD alterations.	16.5 vs. 13.7 (HR: 0.73 in the HRD-positive group, while HRD-negative was closed for futility)	NA
Rucaparib	Ovarian	Maintenance post-ChT	ARIEL3 [206,207] (NCT01968213)	Phase 3 open-label trial vs. physician’s choice, which investigated the efficiency of Rucaparib 600mg bid for up to 24 months as maintenance in 564 patients who responded to primary platinum-based ChT.	10.8 vs. 5.4 (HR: 0.36 in the overall population, regardless of BRCA1/2m or HRD status)	No significant difference in OS in the overall population.
	Prostate	mCRPC monotherapy	TRITON3 [208] (NCT02975934)	Phase 3 open-label trial vs. physician’s choice in patients with mCRPC who progressed after first-line treatment (novel hormonal agent or docetaxel). Patients (405) with BRCA1/2m or ATM alteration were randomized 2:1.	10.2 vs. 6.4 (HR: 0.61 in the overall population) 11.2 vs. 6.4 (HR: 0.50 in the BRCA1/2m population)	NA
Talazoparib	Breast	Metastatic	EMBRACA [209,210] (NCT01945775)	Phase 3 open-label trial, which enrolled 431 patients with gBRCA1/2m advanced breast cancer to compare Talazoparib 1 mg/day to physician’s choice single-agent ChT.	8.6 vs. 5.6 (HR: 0.55)	No significant difference in mOS.
	Prostate	mCRPC (with Enzalutamide)	TALAPRO-2 [211,212] (NCT03395197)	Phase 3 open-label trial, enrolling 805 patients to investigate the efficiency of Talazoparib and Enzalutamide in first-line treatment of mCRPC, regardless of HRD status. Another analysis presented data for the HRD-positive cohort, 399 patients, investigating the efficiency of Talazoparib 1mg/day added to Enzalutamide as first-line therapy of mCRPC.	rPFS: NR vs. 13.8 (HR: 0.45 for HRD-positive) rPFS: NR vs. 21.9 (HR: 0.63 for all participants)	Data are immature.

List of Abbreviations: First line: First-line therapy, ATM: Ataxia-Telangiectasia Mutated, BRCA1/2m: BRCA1/2 mutation, ChT: Chemotherapy, DFS: Disease-Free Survival, gBRCA1/2m: Germline BRCA1/2 mutation, HR: Hazard Ratio, HRD: Homologous Recombination Deficiency, HRR: Homologous Recombination Repair, mCRPC: Metastatic Castration-Resistant Prostate Cancer, mOS: Median Overall Survival, mPFS: Median Progression-Free Survival, NA: Not Available, NR: Not Reached, ORR: Objective Response Rate, OS: Overall Survival, PD: Progressive Disease, rPFS: Radiographic Progression-Free Survival.

**Table 7 ijms-26-03123-t007:** Pivotal clinical trials involving targeted therapies of PIK3CA/AKT1/PTEN alterations and ESR1-mutation.

Biomarker	FDA Approved Therapies	Clinical Trial	Trial Design	Trial Outcomes
PIK3CA activating mutation	Inavolisib + Palbociclib + Fulvestrant	INAVO120 [217] (NCT04191499)	The 325 patients with ER-positive, HER2-negative ABC, who progressed within 12 months after the completion of adjuvant endocrine therapy, were randomized to receive Inavolisib 9mg daily or placebo, with Fulvestrant and Palbociclib.	In the ongoing study, mPFS was improved (15 months vs. 7.3 months, HR 0.43, 95% CI 0.32–0.59 *p* < 0.001), but with higher toxicities associated. Data for OS are still immature.
	Alpelisib + Fulvestrant	SOLAR-1 [218,219] (NCT02437318)	A randomized, Phase 3 trial enrolled 572 patients with ER-positive, HER2-negative ABC, who progressed on previous endocrine therapy.	Improved PFS (11.0 vs. 5.7 months; HR = 0.65, *p* < 0.001) in PIK3CA-mutated tumors. There is no PFS/OS benefit for patients without PIK3CA mutation.
	Capivasertib + Fulvestrant (active on PIK3CA or AKT1 activating mutations, as well as PTEN alterations)	CAPItello-291 [220] (NCT04305496)	The 708 patients with HER2-negative, ER-positive, and ABC who progressed on first-line CDK4/6 inhibitor were enrolled in a Phase 3, randomized, double-blind trial comparing the addition of Capivasertib to Fulvestrant with placebo.	Significant improvement in PFS (7.2 vs. 3.6 months; HR = 0.60, *p* < 0.001); OS data pending.
ESR1 mutation	Elacestrant	EMERALD [221] (NCT03778931)	The 477 patients with ER-positive/HER2-negative ABC, who had 1–2 line(s) of endocrine therapy (CDK4/6 inhibitor, and ≤1 ChT) were randomized in an open-label, Phase III trial to receive Elacestrant 400 mg/day or SoC endocrine monotherapy.	PFS benefit both for ESR1-positive (3.8 vs. 1.9 months; HR = 0.55, *p* < 0.001) and for the general population (2.8 vs. 1.9 months; HR = 0.70); OS trend favoring Elacestrant only for ESR1-positive.

List of abbreviations: ABC—Advanced Breast Cancer. AKT1—AKT Serine/Threonine Kinase 1. CDK4/6—Cyclin-Dependent Kinases 4 and 6. ChT—Chemotherapy. CI—Confidence Interval. ER—Estrogen Receptor. ESR1—Estrogen Receptor 1. HER2—Human Epidermal Growth Factor Receptor 2. HR—Hazard Ratio. mPFS—Median Progression-Free Survival. OS—Overall Survival. PFS—Progression-Free Survival. PIK3CA—Phosphatidylinositol-4,5-Bisphosphate 3-Kinase Catalytic Subunit Alpha. PTEN—Phosphatase and Tensin Homolog. SoC—Standard of Care.

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
