# Peer review of "Next-Generation Sequencing in Oncology—A Guiding Compass for Targeted Therapy and Emerging Applications"

_ijms, 2025, doi:10.3390/ijms26073123_

Round 1

Reviewer 1 Report

Comments and Suggestions for Authors

Review Report for Manuscript:

"Next-Generation Sequencing in Oncology – A Guiding Compass for Targeted Therapy and Emerging Applications"

This review article covers a broad area of literature dedicated to precision medicine. It was undoubtedly a challenge to encompass all the relevant data, but the manuscript requires restructuring to improve readability and logical flow. Currently, it resembles a book chapter rather than a structured review article.    

The aim of the review, stated as:

“Our current review is intended as a broad argument for the value of NGS in various clinical

scenarios and the specific molecular alterations that can be interrogated and specifically

targeted. The proposed clinical scenarios also show the emerging roles of this novel inves-

tigation and provide insights into maximizing its potential. 2. Guiding Compass for Tu-

mor-Agnostic Tumors”

should be clearly positioned at the end of the Introduction section for better readability. The phrase “2. Guiding Compass for Tumor-Agnostic Tumors” seems incomplete or misplaced.

The overall structure of the manuscript should be improved to ensure a smoother flow of information, according to the aim.

Cost-Effectiveness Considerations: The statement: “However, there is growing evidence showing the potential for cost-effectiveness under specific reimbursement policies that would permit adequate access to novel therapies.” highlights an important issue. However, the manuscript lacks a discussion on adequate endpoints for assessing cost-effectiveness in this setting. The authors should address this aspect in more detail.

Abbreviations should be defined at their first appearance (e.g., Page 3, last paragraph). This will improve readability for a broader audience.

Missing Sections: According to the journal's guidelines, a review article should include the following sections: Abstract, Keywords, Introduction, Relevant Sections, Discussion, Conclusions, and Future Directions. Some sections, particularly Future Directions and Discussion, appear to be missing.

The Discussion section should compare NGS with other molecular diagnostic techniques. This would provide a more balanced perspective on its advantages and limitations.

Author Response

We thank the Reviewer for the comments and the patience to review our manuscript. Please find the detailed responses below and the corresponding revisions/corrections highlighted/in track changes in the re-submitted files.

Comment 1: The phrase “2. Guiding Compass for Tumor-Agnostic Tumors” seems incomplete or misplaced.

Answer - Apologies. It was meant to be the headline for the first discussion of this article.  We corrected the issue. 

Comment 2: The overall structure of the manuscript should be improved to ensure a smoother flow of information, according to the aim.

Also in line with the Reviewer's comment: 

Missing Sections: According to the journal's guidelines, a review article should include the following sections: Abstract, Keywords, Introduction, Relevant Sections, Discussion, Conclusions, and Future Directions. Some sections, particularly Future Directions and Discussion, appear to be missing.

The Discussion section should compare NGS with other molecular diagnostic techniques. This would provide a more balanced perspective on its advantages and limitations.

Answer - We corrected the numbering and the re-structured the article. We hope the modification we've made to the article provide a significant improvement to the overall flow of the article, and the Reviewer agrees with these changes. We modified the Discussions chapter and the Future Perspectives to allow for more information and discussion the current and future role and limitations of NGS. We are truly grateful for this opportunity to improve the manuscript!

Comment 3: Cost-Effectiveness Considerations: The statement: “However, there is growing evidence showing the potential for cost-effectiveness under specific reimbursement policies that would permit adequate access to novel therapies.” highlights an important issue. However, the manuscript lacks a discussion on adequate endpoints for assessing cost-effectiveness in this setting. The authors should address this aspect in more detail. 

Answer 3: We added in the Discussions chapter further details on the discussion on Cost-Effectiveness, with valuable references:

"Compelling economic arguments can also include the dramatic decrease in NGS since the early 2000s. Technological advancements and scale economics decreased from $100 million to less than $1,000 in some cases. Unfortunately, costs can still prove prohibitive in low and middle-income countries, especially if a lack of reimbursement means out-of-pocket costs for the patient [237]. This is not to say, however, that NGS analysis and subsequent treatment decisions are not cost-effective, as some systematic reviews have demonstrated financial advantages for NGS-guided treatment decisions [233,238]. Consequently, efforts to develop cost-minimizing workflows for testing, initiatives to develop genomic research capacity, collaborative efforts for genomic data sharing and analysis, and implementation of guidelines mindful of the potential financial toxicity incurred would provide a great benefit in the large-scale adoption of NGS testing [239,240]."

Comment 4: Abbreviations should be defined at their first appearance (e.g., Page 3, last paragraph). This will improve readability for a broader audience.

Answer 4: We searched for missing abbreviations and we corrected and defined them at first appearance. We also modified the abbreviations chapter in our article. We appreciate your suggestion!

Comment 5: The Discussion section should compare NGS with other molecular diagnostic techniques. This would provide a more balanced perspective on its advantages and limitations.

Answer 5: In the Discussion section we added information about the diagnostic accuracy, pitfalls, and advantages of NGS. 

"

The need for significant developments in the precision oncology era requires tools such as NGS for the therapeutic management of oncologic patients. Several reports have provided evidence for the high accuracy diagnostic value of NGS [222,228].

It is important to note, however, that several pitfalls may alter the diagnostic accuracy of NGS, particularly in the pre-analytical phase. As such, the tissue blocks selected for analysis must represent a substantial portion of the tumor (at least 20%) for biomolecular analyses viability [229]. Furthermore, factors such as time to fixation, duration of fixation, and the conditions for tissue storage will significantly affect the nucleic acid integrity of the tumor tissue [230]. Nonetheless, sequencing provides challenges as well, with risks including degradation by RNAses, cross-linking, and fragmentation [231,232]. These pitfalls need to be considered in the evaluation of the NGS workflow, but disadvantages such as long turnaround times, the need for specialized personnel and higher costs are outweighed by the pivotal contribution that NGS brings in oncological management.

As the article written by Zalis et al. remarks, coupling NGS testing with liquid biopsies offers significant advantages, such as an alternative to surgical tissue biopsies in a non-invasive manner (reducing the patient discomfort and risk of complications), monitors cancer progression and response to treatment in real-time (allowing for timely adjustments in therapy), is capable of detecting minimal residual disease, and identifies resistance mechanisms, thus guiding the selection of alternative treatments [228]."

We appreciate your kind remarks, and we hope these changes improved the manuscript! 

Reviewer 2 Report

Comments and Suggestions for Authors

 Nicoleta Galeș et al wrote the review of the impact of NGS  for targeted therapy in oncology. They provided the update and the importance of diagnosis and therapy based on gene mutation and many of them are in clinical trial.

Authors covered a wide area topics that could not be sufficient for a single review. The impact of NGS on diagnosis and therapy of specific cancer could have been more interesting with its limitations as of now and their solutions. They discussed mostly breast and lung cancers and the impact of gene mutations on their diagnosis and therapy.

 This review is a general approach with some information but does not have critically assessed idea to improve the limitations. However, this review has an accumulation of information that could be important for cancer therapy. It has following concerns that are addressed here.

  1. The review does not have line numbers, thus difficult to review.
  2. Authors should include a PRISMA chart or the strategy of literature hunting information as methods.
  3. In page 8, authors mentioned that increase of Her2 leads to expression of 2 million receptor in the cell surface. A cell may not have 2 million proteins, and not at all only receptors.
  4. Authors should include a flow chart/figure showing how each mutation (they described) in EGFR gene for diagnosis and therapy.
  5. Authors completely ignored difficult diagnosed cancers, such as pancreatic, ovarian and brain cancers and the role of NGS to diagnose them. Also no information about various types of blood cancers.
  6. They at least could show a figure depicting the impact (as percent in literature survey) of NGS in successful therapy for various types of cancers.
  7. Authors completely ignored the impact of NGS in detecting tumor antigen which is an important advancement by CAR-T cell therapy

Author Response

We are deeply grateful for the patience and key insights provided by the Reviewer, that have proved most valuable for our manuscript. Please find below the answers to the Reviewer's comments:

Comment 1: The review does not have line numbers, thus difficult to review.

Answer to comment 1 - Apologies. We verified our version, and edited. We hope this issue does not repeat. 

Comment 2: Authors should include a PRISMA chart or the strategy of literature hunting information as methods.

Answer to comment 2 - The review is intended to be narrative. No systematic searches have been made in the literature. For full disclosure, the starting point of our article were the current guidelines published by oncology societies such as NCCN, ASCO, and ESMO. Starting from there we showcased the importance of NGS as an ubiquitous tool synonymous with precision oncology in the modern era, using the data from the relevant literature and trials that have lead to the approval of therapies and indications in the guidelines. As we stated, the review is non-systematic, narrative, and the report retrieval aspect is not indicated in this scenario.

Comment 3 - In page 8, authors mentioned that increase of Her2 leads to expression of 2 million receptor in the cell surface. A cell may not have 2 million proteins, and not at all only receptors.

Answer -We have changed the sentence in order to not cause any confusion:

"Overexpression of the Her2-neu gene leads to a 40-100-fold increase in HER2 protein, which in turn leads to overexpression on the cellular surface"

Comment 4:  Authors should include a flow chart/figure showing how each mutation (they described) in EGFR gene for diagnosis and therapy.

Answer - Figure 1 was added. We are thankful for this idea as it really helped us provide a better overview for the oncogenic-driven therapies in NSLCC. 

Comment 5: Authors completely ignored difficult diagnosed cancers, such as pancreatic, ovarian and brain cancers and the role of NGS to diagnose them. Also no information about various types of blood cancers.

Answer - Section 2.1. was added, in order to contribute to an existing paragraph about the paradigm-shift in glioma classification brought by molecular profiling. Ovarian cancer is thoroughly discussed in the 2.4. section of our manuscript. We are focusing mostly on therapies which have the clearance and consensus from international oncology guidelines (such as NCCN, ASCO, ESMO) and use NGS in order to facilitate the treatment. For pancreatic cancer most of the discussion would revolve around the same therapies approved for tumor-agnostic therapies, discussed in section 2.2.

Also, this article is intended for discussing NGS impact on solid tumor oncology, as we have mentioned, and we have also now underlined this aspect. Our expertise does not include blood cancers, and as such, we consider an attempt to provide an insight on this topic as being disingenuous.

If there are further suggestions, we are welcoming them with great interest! 

Comment 6: They at least could show a figure depicting the impact (as percent in literature survey) of NGS in successful therapy for various types of cancers.

Answer - Apologies, this suggestion is a bit confusing. Firstly, we are wondering how we could come up with a percent in literature survey for the impact of NGS. We can only imagine that such a percent would come as a result of a meta-analysis. However, this article is a narrative review, and not a meta-analysis. As such, it's scope is to present the current data and structure the information in a manner which is more accessible for the reader. In order to argument that NGS is impactful, we provided the data from key clinical trials. Furthermore, we referenced impactful research that used meta-analyses in providing such data. However, this field is ever-changing and a figure showing a statistic of the impact of NGS would only represent a snapshot of an epheremal status quo. Surely, the Reviewer can expand on this comment and we are more than welcoming of any suggestions that could satisfy this request. 

Comment 7: Authors completely ignored the impact of NGS in detecting tumor antigen which is an important advancement by CAR-T cell therapy

Answer - Unfortunately, we did, and we apologize. As mentioned before our expertise does not include blood cancers, and CAR-T cell therapy in the treatment of solid tumors is an ongoing struggle. We included a paragraph in the Future Directions section discussing afamitresgene autoleucel as a new therapy for synovial sarcomas:

"Our narrative review showcases the benefits of NGS in guiding diagnosis, targeted therapy, and immune therapy by interrogating multiple complex genomic alterations. The development of new therapies, such as CAR-T cell therapy for solid tumors, may result in new opportunities for NGS adoption. Trials such as the SPEARHEAD-1 in synovial sarcomas show promising results for the adoption of CAR-T cell therapies such as afamitresgene autoleucel and require molecular profiling for melanoma-associated antigen A4 (MAGE-A4) expression and HLA Typing [241]. "

We are honored by these kind suggestions and we hope we have provided substantial amendments to our article, that satisfy the Reviewer's high standard for publishing! 

Reviewer 3 Report

Comments and Suggestions for Authors

Here the authors reviews the significance of Next-Generation Sequencing (NGS) technologies in the field of oncology. They emphasize the role of NGS in identifying actionable alterations and facilitating the development of targeted therapies that significantly improve clinical outcomes. The review highlights the pivotal role of molecular guidance in treatment decisions and showcases why NGS is crucial in modern oncology. The use of NGS has led to advancements in precision oncology, enabling comprehensive genomic evaluation, and identifying biomarkers crucial for targeted therapy. THis review is within the readership of the journal,  and comprehensive.

Major points

The article provides an in-depth analysis of the various NGS assays and their applications in oncology, offering a detailed look into how these technologies aid in precision medicine.

It emphasizes the real-world clinical applications of NGS, illustrating its impact on treatment decisions and outcomes in cancer therapy.

They discuss  emerging applications and the future potential of NGS in multicancer early detection tests and tumor-agnostic therapies showcases forward-thinking perspectives.

By including data from key clinical trials, the article provides evidence-based insights into the efficacy and safety of targeted therapies guided by NGS.

They highlight the importance of Molecular Tumor Boards in integrating NGS results into clinical practice, promoting a collaborative approach to cancer treatment.

Minor points 

While the article mentions the cost-effectiveness of NGS under specific policies, it should delve deeper into the economic challenges and solutions for broader implementation.

The article should explore more on the disparities in access to NGS technologies and the implications for patients in low-resource settings.

A few sentences on the role of NGS in targeted/ degrader-based therapies would be interesting to add, cf Targeted next generation sequencing as a tool for precision medicine | BMC Medical Genomics | Full Text, Novel Therapeutic Approaches Targeting Post-Translational Modifications in Lung Cancer, Immunotherapy and next-generation sequencing guided therapy for precision oncology: What have we learnt and what does the future hold? - PMC

Author Response

We are honored by the kind comments and insightful suggestions of the Reviewer. Please allow us to respond to the comments. We are convinced that the necessary changes we have made to the manuscript as a result of the Reviewer's advice significantly improved the quality of this article, and we hope the Reviewer finds them acceptable. 

Comment 1 - While the article mentions the cost-effectiveness of NGS under specific policies, it should delve deeper into the economic challenges and solutions for broader implementation. The article should explore more on the disparities in access to NGS technologies and the implications for patients in low-resource settings.

Answer 1 - Excellent idea, we have modified the Discussion section of our article to provide an insight in the economics and real-life implementation of NGS, with insightful literature on the topic. 

"Compelling economic arguments can also include the dramatic decrease in NGS since the early 2000s. Technological advancements and scale economics decreased from $100 million to less than $1,000 in some cases. Unfortunately, costs can still prove prohibitive in low and middle-income countries, especially if a lack of reimbursement means out-of-pocket costs for the patient [237]. This is not to say, however, that NGS analysis and subsequent treatment decisions are not cost-effective, as some systematic reviews have demonstrated financial advantages for NGS-guided treatment decisions [233,238]. Consequently, efforts to develop cost-minimizing workflows for testing, initiatives to develop genomic research capacity, collaborative efforts for genomic data sharing and analysis, and implementation of guidelines mindful of the potential financial toxicity incurred would provide a great benefit in the large-scale adoption of NGS testing [239,240]."

Comment 2 - A few sentences on the role of NGS in targeted/ degrader-based therapies would be interesting to add, cf Targeted next generation sequencing as a tool for precision medicine | BMC Medical Genomics | Full Text, Novel Therapeutic Approaches Targeting Post-Translational Modifications in Lung Cancer, Immunotherapy and next-generation sequencing guided therapy for precision oncology: What have we learnt and what does the future hold? - PMC

Answer 2 - Very insightful comment, we provided further details in the Future Directions section and we are grateful for the opportunity to expand the precision oncology argument towards pharmacogenetics. 

"Furthermore, the considerable impact of NGS in interrogating for polymorphisms of the same gene can be greatly useful in identifying genetic variations of pharmacogenes, implying a near future where therapy toxicity can also be adjusted according to the individual characteristics of each patient [242]. It is, therefore, highly likely that NGS will become synonymous with the practice of precision oncology in the management of solid tumors."

We once again want to extend our gratefulness towards the Reviewer's patience and we hope the modifications satisfy the Reviewer's standards! 

Round 2

Reviewer 1 Report

Comments and Suggestions for Authors

The authors have adequately addressed my comments, and the manuscript can be accepted for publication.